# A qualitative description of HIV testing and healthcare experiences among trans women in Ghanaian urban slums BSGH-012

Henry Delali Dakpui[1], Osman Wumpini Shamrock[1,2]*, Gloria Aidoo-Frimpong[3,4], Edem Yaw Zigah[1,2], George Rudolph Kofi Agbemedu[1], Abdallah Ahmed[5], Umar Osman[3], Aliyu Haruna[2], Gamji Rabiu Abu-Ba'are[1,2,5,6]

**1** Behavioral, Sexual, and Global Health Lab, North Legon, Accra, Ghana, **2** Behavioral, Sexual, and Global Health Lab, School of Nursing, University of Rochester, Rochester, New York, United States of America, **3** Center for Interdisciplinary Research on AIDS, School of Public Health, Yale University, New Haven, Connecticut, United States of America, **4** Department of Epidemiology and Environmental Health, University at Buffalo, Buffalo, New York, United States of America, **5** Hope Alliance Foundation, Accra, Ghana, **6** Department of Public Health Sciences, University of Rochester, Rochester, New York, United States of America

* osmanwumpini_shamrock@urmc.rochester.edu

**Data Availability Statement:** All data can be found in the manuscript and Supporting information files.

## Abstract

Achieving the UNAIDS 90–90–90 targets hinges on identifying and engaging individuals with HIV in care, requiring 90% of those infected to be diagnosed, initiated on ART, and achieving viral suppression. Despite this imperative, HIV testing services as well as research in Ghana often overlook the unique experiences of transgender women in urban slums, impacting their engagement with care. Using the gender affirmative model lens, this study reports the HIV testing experiences of trans women in Ghanaian slums, highlighting how the healthcare environment, counseling, and healthcare provider attitudes shape these experiences. The study used a qualitative descriptive interview design with 20 trans women aged 18 to 31 from urban slums in Greater Accra Metropolitan Area, Ghana. Participants were recruited through purposive and snowball sampling. Data were collected via face-to-face interviews, transcribed, and analyzed using NVivo, with results presented in categories and subcategories. Two main categories emerged from our data analysis: 1) Positive Experiences with HIV Testing, and 2) Negative Experiences with HIV Testing. Positive experiences with HIV testing among trans women in Ghanaian slums included a welcoming environment at healthcare facilities, supportive counseling, and relatability with HIV-positive nurses. Negative experiences were characterized by fear and anxiety during testing, often intensified by healthcare worker attitudes, including unwelcoming behaviors and judgmental body language, especially in facilities that are not key population friendly. These categories provided a framework for understanding the varied experiences of trans women in Ghanaian slums regarding HIV testing. The study highlights the urgent need for Ghanaian slum healthcare facilities to address discrimination against trans women by creating inclusive, supportive environments. It stresses the importance of using a gender-affirmative approach to improve HIV testing and health outcomes for trans women. Policymakers and healthcare

**Funding:** The authors received no specific funding for this work.

**Competing interests:** The authors have declared that no competing interests exist.

providers must focus on training, inclusive care, and cultural competence to reduce health disparities for this population.

## Introduction

In low- to middle-income countries, transgender women (TGW) are 77.5 times more likely to have HIV than cisgender women [1]. Ghana, a lower-middle-income country in West Africa, faces a concentrated HIV epidemic among key populations [2]. In 2017, an estimated 28.1% of TGW in Ghana were living with HIV, a prevalence rate exceeding that of all other key populations in the country and far surpassing the national average of 1.7% [3].

While HIV testing has been firmly established as a key and critical step in curbing the epidemic, the existing HIV testing coverage and frequency pose a barrier [4–11]. In low-and middle-income countries, less than half of people living with HIV are aware of their status [12]. In Ghana, despite a national increase in HIV status awareness from 43% in 2014 to 72% in 2023, key populations, of which TGW are a part, continue to experience low rates of HIV testing due to pervasive stigma and discrimination that permeate various aspects of their lives [3, 7, 8, 13–26].

Within the sub-Saharan region of Africa, TGW continue to face severe societal stigmatization, reinforced by both cultural norms and legal structures [27, 28]. Cultural norms in many sub-Saharan African societies often dictate rigid gender roles and expectations, leading to the marginalization of individuals whose gender identity do not conform to the dominant traditional norms [29, 30]. Similar to other African nations, Ghana is deeply religious, which correlates with widespread anti-LGBT (lesbian, gay, bisexual, and transgender) sentiment [20]. Many Ghanaians perceive non-heteronormative sexual orientations and gender identities as "un-African" or as Western imports, rather than recognizing them as human rights issues [31]. As a result, physical and violent homophobic attacks against LGBT individuals are frequent and often endorsed by the media, as well as religious and political figures [22, 32].

Additionally, legal structures in several sub-Saharan African countries, including Ghana, criminalize same-sex relationships and behaviors, contributing to the reinforcement of stigma against transgender individuals [33, 34]. In Ghana, a recently passed anti-LGBT bill by parliament is pending presidential assent to become law, with significant international and domestic pressure urging the president to reject this discriminatory legislation [35]. The bill, otherwise known as the "Promotion of proper human sexual rights and Ghanaian Family Values Bill," imposes prison sentences for identifying as LGBT and forming or funding LGBT groups, with penalties of up to three years for individuals and up to five years for groups [35, 36]. The bill also proposes harsh measures like jail terms of up to 10 years for LGBT advocacy campaigns aimed at children and encourages the public to report LGBT individuals to authorities for action. These legal frameworks not only seemingly perpetuate discrimination but also create barriers to accessing essential healthcare services and support for TGW. The criminalization of homosexuality in Ghana further intensifies societal prejudices and biases against LGBT individuals, including TGW [33, 34].

Moreover, within the Ghanaian context, HIV stigma remains profoundly ingrained and fueled by misconceptions, fear, limited knowledge, and the association of HIV with sin and death [37]. Persons living with HIV often endure rejection, insults, violence, and sometimes denial of health care, resulting in social isolation, restricted healthcare access, and reluctance to seek treatment, ultimately exacerbating the HIV epidemic [37].

In healthcare settings, pervasive stigma and discrimination against TGW in Ghana persist, attributed to factors such as a lack of cultural competence among healthcare workers, absence of LGBTQI-specific training in the healthcare sector, and ingrained societal prejudices [20, 38]. Many healthcare providers in sub-Saharan Africa lack the understanding and sensitivity required to address the unique health needs of TGW, often leading to discriminatory practices or reluctance to provide appropriate care [7, 39]. For instance, research across Kenya, Malawi, and South Africa revealed that nearly half (45.3%) of the men who have sex with men (MSM) and TGW reported at least one healthcare-related stigma experience [40]. In South Africa, transgender individuals encountered stigma while accessing reproductive healthcare, while other transfeminine and gender-diverse women faced discrimination due to their gender identity and expression [41]. They described hostile health services, with unrelated health issues often linked to gender or sexual identity, causing discomfort. In Uganda, TGW experienced various forms of stigma, including police and client violence, workplace and familial discrimination, and a lack of tailored health services [42].

These challenges faced by key populations are exacerbated, especially for those residing in urban areas, particularly within slum communities [14, 43], as residing in these areas, further increases the risk of HIV infection among this population [14, 44]. Urban slums are informal settlement and pose a multitude of social and structural challenges that amplify vulnerability to HIV, which include crowded living conditions, discrimination, poverty, limited HIV knowledge, high rates of transactional sex, and limited healthcare access [45]. Notably, certain slum areas within Ghana's capital, Accra, such as Agbogboloshie (5%), Okai Koi North (8%), and Madina (7%), report a higher HIV prevalence, averaging 7% compared to the national prevalence of 1.7% [13, 46]. TGW in these areas are likely to have low HIV testing rates due to poor access to services, low HIV risk perception, lower education levels, and poverty [47, 48].

Additionally, in low-resource settings, the impact of stigma on healthcare access is even more pronounced, presenting a significant barrier to service utilization compared to other contexts [49]. Healthcare-related stigma not only leads to avoidance of healthcare services but also adds discomfort and stress, adversely affecting health outcomes [14, 50]. For instance, in Ghana, intersectional stigma and discrimination experienced by key populations not only deters them from seeking healthcare services, resulting in delayed diagnosis and treatment of physical health conditions, but also contributes to increased rates of depression, anxiety, and other mental health issues, as well as social exclusion and isolation [15, 51]. Furthermore, stigma and discrimination experienced by transgender individuals in other low-income settings have been linked to adverse health effects, including depressive symptoms, anxiety, suicidality, substance abuse, condomless sex and increased HIV transmission [52].

Research specifically focused on TGW, especially those living in slum communities, in Ghana is limited, with most studies focusing on gay, bisexual, other men who have sex with men, and female sex workers [14, 15]. This paper seeks to address this knowledge gap by offering a qualitative exploration of HIV testing and care experiences within the context of TGW living in Ghanaian urban slums, which can have significant implications for public health interventions and policy development. Identifying the unique challenges and barriers encountered by TGW can contribute to the design of more inclusive and practical strategies to combat the spread of HIV in this vulnerable population.

Grounded in the gender-affirmative model, this study aims to explore the nuanced landscape of HIV testing experiences among TGW living in Ghanaian slums, emphasizing both the supportive and challenging aspects of their experiences [53]. The gender-affirmative model prioritizes the affirmation of gender identity, empowerment, and the mitigation of stigma and discrimination. By applying this framework, we emphasize both the supportive and challenging aspects of HIV testing for TGW, aiming to understand how these experiences influence

their access to healthcare [54]. This framework is particularly relevant to our objective of expanding access to HIV testing services. By understanding the facilitators and barriers TGW face, we aim to provide insights that can help policymakers, healthcare providers, and community organizations design services that are not only more accessible but also more welcoming and affirming of TGW's identities. The ultimate goal is to ensure that HIV testing services are tailored to meet the specific needs of TGW, thereby improving their healthcare outcomes in Ghanaian slums.

## Methods

### Research design

We employed a qualitative descriptive interview design to explore HIV testing and healthcare experiences among 20 TGW living in urban slums in the Greater Accra Metropolitan Area of Ghana. This design was selected for its suitability in capturing the complex, nuanced experiences of TGW in urban slums—a population whose voices are seldom heard in traditional research [55].

### Sampling and recruitment procedure

Participants were recruited using purposive and snowball sampling techniques, leveraging existing relationships with key populations. Research assistants from our partner community organization, who had established connections within the TGW community, initiated the process. A map of existing slum communities in Accra was used to identify three different slums from which the initial participants were recruited.

Four TGW from these communities were initially identified and recruited. Each of these initial participants then referred additional TGW from their social networks, leading to a snowball effect: the first four referred six, those six referred one each, and another six were subsequently referred. In total, 22 TGW were recruited, but only 20 met the inclusion criteria after completing a prior questionnaire designed to confirm their eligibility. Participants were compensated GHC 200 ($17) to cover transportation costs.

### Inclusion and exclusion criteria

Eligibility criteria for participation required TGW to reside in an urban slum in Accra, self-identify as transgender, be aged 18 or older and have engaged in sexual activity with a male within the past year. Individuals not biologically born male or residing outside urban slum communities in the Greater Accra Metropolitan Area were excluded from the study.

### Data collection procedure

Data collection involved face-to-face interviews with TGW participants, and were collected between 30th May 2023 and 30th June 2023. Before the interviews, eligible individuals were provided with information sheets detailing the study and given a choice to sign consent forms, with research assistants offering additional clarification and reminders about consent during the process. All interviews occurred in secure community partner locations, with nine out of 20 conducted in Twi, a local Ghanaian language, and the remainder in English. The sample size of 20 participants was determined to be sufficient for this qualitative study as data saturation was achieved, indicating that no new significant themes was identified after the 15th interview, thus ensuring comprehensive coverage of the experiences within the target population [56].

## Nature of questions

To ensure a comprehensive exploration of HIV testing experiences among TGW in urban slums, an interview guide was developed collaboratively with experts in urban slum communities research, healthcare practice, and transgender engagement. The guide included open-ended questions and prompts aimed at eliciting narratives about positive and negative experiences encountered during HIV testing. Sample questions included: When last did you test for HIV in your community? What were your experiences like during your last HIV test? What positive experiences have you had at HIV testing sites in your community? Could you describe any challenges or negative experiences you've faced while seeking HIV testing services in your community? In addition to the questions, the guide also included a semi-structured script designed to support the interviewers. This script provided standardized introductory statements, instructions, and closing remarks, ensuring consistency across interviews.

## Analytical strategy

Interviews were recorded and transcribed verbatim by research assistants, with translations from Twi to English undertaken by experienced team members who had previously performed such translations in prior studies [14]. Data analysis was conducted using NVivo, employing a systematic approach to ensure the rigor of the findings. Initially, a team of six research assistants developed a comprehensive codebook, which was refined collaboratively, and the first five transcripts were coded collectively. Subsequently, the remaining transcripts were divided, with two team members independently coding each transcript to ensure inter-coder reliability and capture nuances in the data. Coding comparisons were utilized to maintain consistency and validate decisions, with any discrepancies resolved through team discussions. This iterative process ensured the reliability and validity of the analytical findings. Throughout, diligent efforts were made to de-identify the data during transcription to adhere to ethical guidelines. After careful data comparison in NVivo, key themes representing main patterns and significant findings were identified. All monetary amounts quoted were converted using the prevailing exchange rate at the time of data collection (US$1 = GH ₵12).

## Ethical approval and informed consent

Ethical approval for this study was obtained from the Research Subjects Review Board (RSRB) at the University of Rochester University (STUDY00008151) and the Ghana Health Service Ethics Review Committee (GHS-ERC:002/03/23). In addressing the ethical considerations specific to the vulnerable population of TGW in urban slums, our study implemented several protective measures. Recognizing the heightened risks of stigma and discrimination, we ensured that all participant interactions, from recruitment through interviews, were conducted with utmost confidentiality and respect. Inform consent was sort from all participant before data collection commenced. We also employed stringent data de-identification techniques and secure data handling practices to safeguard participant identities. Interviews were held in secure, private locations chosen by participants to ensure their comfort and safety. Additionally, research assistants were specially trained in sensitive communication and handling potentially distressing topics, ensuring that discussions did not exacerbate participants' vulnerabilities. These measures collectively aimed to uphold the dignity and safety of participants while collecting vital data on their HIV testing experiences.

## Results

### Description of participants

The participants in our study consisted of 20 TGW with a mean age of 20 years, ranging from 18 to 31. Educational backgrounds varied, with 10 participants having completed secondary education, four with tertiary education, five with primary education, and one without any formal education. All were employed, earning an average monthly income of GH₵830.15 ($69.2), with incomes ranging from GH₵200 ($16.7) to GH₵2003 ($166.91). In terms of religious affiliation, the majority were religious: 15 identified as Christians, four were non-religious, and one practiced Islam. A reflection of the high religiosity in the Ghana [57–59]. Regarding sexual behavior, 19 participants reported having exclusively male sexual partners over the past year, and one reported having partners of both sexes. Although the study did not explicitly inquire about HIV status to avoid potential stigma, three participants voluntarily disclosed that they were living with HIV.

### Description of categories and subcategories

As shown in Table 1, two main categories emerged: Positive Experiences with HIV Testing, and Negative Experiences with HIV Testing. Under Positive Experiences, subcategories included; Welcoming Environment at healthcare facilities, Counseling and Relatability with HIV-positive nurses, and Inclusive, Informative, and Supportive Counseling. Negative Experiences encompassed Fear and Anxiety in HIV Testing and Negative Health Worker Attitude. These categories provide a framework for understanding the varied experiences of TGW in Ghanaian slums regarding HIV testing.

### Positive experiences with HIV testing

**Welcoming environment at healthcare facilities.** Some participants reported feeling welcomed and cared for by healthcare staff during HIV testing. One participant highlighted the reassuring presence of a nurse who provided counseling while waiting for test results. The nurse's supportive words, delivered by a white nurse as noted by the participant, significantly alleviated the participant's anxiety and instilled confidence in managing a potential reactive outcome.

> They are so welcoming. They are so nice. Yeah. And they are good too. Like they take time to take care of us. When you go there, they have patience. I met this nurse. She's White though. And then she was very welcoming. She started counseling with me while my results were pending. I was nervous and scared. But when I walked in, she was like, baby keep

**Table 1. Summary of positive and negative HIV testing experiences reported by trans women in Ghanaian slums.**

| Category | Subcategory | Description |
|---|---|---|
| Positive Experiences with HIV Testing | Welcoming Environment at Healthcare Facilities | Participants felt welcomed and cared for by healthcare staff. |
| | Counseling and Relatability with HIV-Positive Nurses | Nurses provided personalized counseling, shared their own experiences, and reduced stigma. |
| | Inclusive, Informative, and Supportive Counseling | Nurses engaged in comprehensive discussions, covering HIV, PrEP, and safe sexual practices. |
| Negative Experiences with HIV Testing | Fear and Anxiety in HIV Testing | Participants experienced fear and anxiety, particularly during delays or multiple retests. |
| | Negative Health Worker Attitude | Participants encountered unwelcoming behaviors and judgmental attitudes from healthcare workers. |

calm, relax, there's nothing to be afraid of. When you're reactive, there are ways to go by to live as long as you're supposed to live. And so, like her words actually calm me down and like, I am in for this and I'm okay.

—Participant 6 (23, Christian)

**Counseling and relatability with HIV positive nurse.**   Another participant shared an encounter with a nurse who went beyond standard testing procedure by discussing the importance of medication adherence and a healthy lifestyle. The nurse even disclosed her own HIV-positive status, aiming to reduce stigma and emphasize that with proper care, individuals with HIV can lead fulfilling lives.

Before I began my medications, they conducted the tests again. One of the nurses took me to a different room and told me that for HIV, if you take your medications, it won't progress. Make sure to take your medications on time, eat healthy, and drink lots of water. This is what will keep you healthy. If you miss your medications or fail to eat well, it can bring your system down. She asked if I would believe her if she told me she had HIV, and I said no. What shows she has HIV? She revealed her status as positive to me, but I didn't believe it because she looked very fine. She is a very nice woman without any visible signs of HIV. So, when it was time to check the viral load, she did it on herself before and her results were positive. This calmed me down greatly as I realized that if you have HIV, you are not rejected in society because HIV is better than some diseases. As long as you take your medications, no one would know.

—Participant 2 (19, Christian)

**Inclusive, informative and supportive counseling.**   Participants shared experiences where nurses engaged in comprehensive discussions that went beyond the technical aspects of HIV testing. This included education on HIV, PrEP, and safe sexual practices, providing a holistic approach to healthcare

I went there. I met a nurse and then she asked me my sexual orientation, my gender identity, and then she educated me on HIV and PrEP and then told me the outcome. . . um, whatever was going to come. So, we did the testing and while waiting for the results, she engaged me in conversations about safe sexual relationships. So, the results came and then I was fine.

—Participant 7 (26, Christian)

Another participant shared a positive account of their HIV testing experience at the health facility. The participant expressed that the atmosphere during the HIV test was serene and comfortable. The participant recounted the supportive response they received upon learning of their positive test result. The nurse provided consolation and supportive counseling, establishing a human connection beyond the clinical aspects of the test.

When I went to test for HIV It was serene it was okay the nurse was cool with me, they did not do anything to make me feel like not going there again. because they took me through counseling first, after they counseled me and then like, we had a friendly chat. And then I did the test and after the test came out positive the nurse consoled me, and we still talk

because he took my number then he will be checking on me and then when the time is due for me to take my medication, they let me know then I go for it. As for me I see them to open their door for us to come in whilst to get to check our health status without any problem.

—Participant 4 (22, Christian)

## Negative experiences with HIV testing

**Fear, and anxiety in HIV testing.**   Fear and anxiety were prevalent among participants, especially when faced with delays or the need for multiple retests. These encounters led them to speculate about receiving a positive diagnosis, intensifying their apprehension.

I was very terrified. We were there, they took the test and my result wasn't coming. It was delayed and they said I had to redo the test. They did it and they were like they were not getting the result, so I had to do it again for the third time and at that point, I was scared the test was positive, and the health workers didn't want to tell me the truth. At that point, I was like, okay, let me accept the fact that I am positive till the result came out that they wanted to confirm because the first one wasn't successful, the second one was negative and they wanted to confirm it was negative so that is why they did the third one.

—Participant 3 (18, Christian)

Participants also cited the influence of witnessing others' emotional reactions to positive diagnoses as a source of communal anxiety. This communal anxiety heightened their own apprehension about undergoing testing, as expressed by another participant.

My experience was with the people around. Some were scared of being tested. I was also scared of being tested. Because you know, when someone has it, they come out and then cry. It's very sad, but you can't do anything about it, it has happened. So, all you need is to console the person and say something sensible to the person so that the person will be okay.

—Participant 1 (26, Christian)

Additionally, participants pointed out that interactions with healthcare providers, including intrusive questioning, often contributed to the fear and apprehension they felt during the testing process.

In the health facilities, sometimes they put fear in you. The nurses put fear in you. Like, ask you what have you been doing and all that. So, when the test is run, sometimes you sit down to think about the people you have sex with, looking at their bodies and wondering if they might be having HIV

—Participant 8 (31, Christian)

**Negative health worker attitude.**   Participants highlighted instances of encountering negative attitudes from healthcare staff, especially in facilities that were less inclusive of key populations. Trans women often felt unwelcome due to their feminine appearance, encountering unwelcoming behaviors such as awkward stares and negative body language. In these

unwelcoming hospitals, nurses would sometimes gossip and judge them, leading to a hostile environment.

> Mostly when I visit a health facility which is not key population oriented for testing, the workers there make you feel like you shouldn't be attended to because of the way you appear looking feminine. Sometimes too, their body language, you know, when you are at the facility, they can look at you from top to bottom. Without saying anything, their body language is speaking to you negatively.
>
> —Participant 12 (24, Christian)

Despite the usual process of receiving care in the Outpatient Department, these participants had to assert themselves to be attended to.

> There was an instance when I went to the hospital that was not key population friendly for testing, and all of a sudden, everyone was looking at me, because of the way I dressed. Everyone was looking. The nurses started gossiping. I sat there for about one hour until I approach them to tell them why I was there. But on a normal basis, you know, you go through the Out Patient Department, and a nurses will attend to you. In my case no one wanted to attend to me due to how I looked and identified myself. I had to approach them before I was attended to.
>
> —Participant 10 (25, Muslim).

## Discussion

The findings of this study offer valuable insights into the experiences of TGW in Ghanaian slums regarding HIV testing, contextualized within the gender-affirmative model [54]. Our findings resonate with the principles of the gender-affirmative model, emphasizing the significance of the healthcare environment, counseling, and healthcare provider attitudes in shaping the HIV testing experiences of TGW.

Consistent with existing literature and the principles of the gender-affirmative model, which emphasizes affirmation of gender identity, empowerment, and mitigation of stigma and discrimination, our findings highlight the importance of creating a welcoming environment within healthcare settings [54, 60, 61]. A welcoming environment serves as a practical application of these principles by affirming and validating the gender identities of TGW, alleviating anxiety, building confidence, and fostering trust during the HIV testing process, thereby encouraging regular healthcare engagement, which is essential for continuous HIV prevention and treatment [62, 63]. The presence of supportive and empathetic healthcare providers, as mentioned by some TGW in this study has also been shown to foster trust and encourage individuals to seek HIV testing, thereby promoting HIV prevention and care [62, 64].

The significance of counseling and peer support in healthcare settings has been widely recognized in the literature as a key factor in enhancing the HIV testing experiences of TGW and MSM [65]. Our findings corroborate this, emphasizing the importance of counseling and relatability in reducing stigma, providing emotional support, and promoting positive health outcomes. Sharing personal experiences, as demonstrated by the HIV-positive nurse in our study, can be a valuable strategy for reducing the social stigma surrounding HIV and promoting medication adherence and healthy lifestyle choices among people living with HIV. By integrating counseling and peer support within a gender-affirmative framework, healthcare

providers can create a supportive and affirming environment that addresses the unique needs and challenges faced by TGW in the context of HIV testing and care. This approach not only enhances the accessibility and acceptability of HIV testing services but also contributes to improving health outcomes and reducing health disparities among this population.

Our findings further highlight the value of inclusive, informative, and supportive counseling in creating a positive healthcare experience among TGW in Ghanaian slums. By fostering an environment that prioritizes affirmation of gender identity, empowerment, and the mitigation of stigma and discrimination, healthcare providers who engage in comprehensive discussions about sexual orientation, gender identity, and provide education on HIV and PrEP contribute to a more positive and empowering HIV testing experience for TGW [66, 67]. This approach creates trust, builds rapport, and encourages open communication, promoting improved health outcomes and reducing health disparities in this population [54]. In contrast to studies conducted in other parts of Africa, where TGW often face significant barriers and negative experiences in healthcare settings, our findings suggest that a supportive and affirming approach can lead to more positive experiences [68].

In this study, we also found that fear and anxiety during HIV testing process, negative attitudes, and discriminatory practices of nurses were prevalent factors characterizing the negative experiences of TGW in Ghanaian slums. Fear and anxiety during the HIV testing process, as reported by the participants in our study, have been identified in previous research as common experiences across various populations and found to be significant barriers to HIV testing [68, 69]. These findings underscore the need for healthcare facilities to improve their testing procedures and communication strategies to minimize anxiety and fear. Effective communication, transparency about the testing process, and timely delivery of results are essential to alleviate anxiety and promote a more positive testing experience [69].

Furthermore, negative attitudes and discriminatory practices within healthcare settings have been consistently reported by TGW and are considered a significant barrier to accessing healthcare services, including HIV testing, among TGW [70, 71]. These negative experiences can lead to avoidance of healthcare services, delayed HIV testing, and reduced engagement in HIV care, which ultimately contribute to health disparities and poorer health outcomes among TGW [71, 72]. In line with the gender affirmative model, these findings highlight the urgent need for healthcare facilities to address and eliminate discriminatory practices and attitudes toward TGW [54]. Healthcare providers must receive comprehensive training on key population health issues, cultural competence, and respectful communication to ensure equitable and inclusive care and promote a more positive and welcoming healthcare experience for TGW.

Notably, participants in our study reported experiencing negative attitudes and discrimination specifically within non-key population (KP) inclusive facilities. These findings suggest that the type of facility—whether it is KP-inclusive or not—can significantly influence the quality of care received. KP-friendly healthcare facilities are extremely limited in Ghana and mostly private owned. Private facilities like the West Africa AIDS Foundation (WAAF) and the International Health Clinic Center (IHCC) have been recognized for their KP-inclusive approaches, which contrast sharply with the negative experiences reported in more general healthcare settings [73]. These facilities provide a model of care that could be emulated by public healthcare facilities across Ghana to improve the inclusivity and quality of care for not just TGW but key population in general [73].

Furthermore, the advent of HIV self-testing (HIVST) and its introduction in Ghana presents a promising solution for addressing some of the barriers identified [74]. HIVST allows individuals to perform an HIV test in the privacy of their own homes, reducing the immediate need to engage with potentially discriminatory healthcare settings [75]. This approach can

help mitigate the impact of negative attitudes and discriminatory practices by offering a less stigmatizing and more confidential testing option. HIVST can empower TGW by providing them with greater control over their testing experience and reducing the reliance on healthcare settings where they might face stigma. By improving access to testing while avoiding direct interaction with healthcare staff who may hold negative attitudes, HIVST aligns with the principles of the gender-affirmative model by enhancing empowerment and reducing stigma [59]. Additionally, as awareness and accessibility of HIVST increase, it could encourage more TGW to seek testing and engage with healthcare services, ultimately leading to better health outcomes and reduced health disparities.

## Strengths and limitations

This study notably contributes to the field by focusing on the often-underrepresented HIV testing experiences of TGW in urban slums within Ghana, employing the gender-affirmative model to validate and respect gender identities. The qualitative interview approach allowed for an in-depth exploration of the personal and social nuances associated with HIV testing, which quantitative methods might fail to capture. Additionally, the collaboration with local community organizations not only facilitated access to this hard-to-reach population but also ensured that our methodology was culturally sensitive and tailored to the specific needs of the participants. However, the study has limitations. It does not fully explore the intersectional factors such as ethnicity, socioeconomic status, and education level that might influence HIV testing experiences and outcomes. To address this, we acknowledge these gaps and recommend them as focal points for future research, and we have initiated discussions with community partners to integrate some of these considerations in ongoing programs. The geographical scope is also limited to the Greater Accra Metropolitan Area, which may restrict the generalizability of the findings. Future research should aim to include other regions to enhance the breadth of data and applicability of the findings. Additionally, the use of snowball sampling and a relatively small sample size may introduce selection bias and limit the diversity within the sample. Despite these potential biases, we utilized rigorous qualitative methods to ensure the reliability of our findings and plan to employ more diverse sampling techniques in future studies to improve the representativeness of the sample.

## Conclusion

This study contributes to the literature on the HIV testing experiences of TGW in Ghanaian slums and highlights the need for healthcare facilities to address discriminatory practices and attitudes towards TGW and create a welcoming and affirming environment within healthcare settings to enhance HIV testing experiences and promote improved health outcomes and reduced health disparities in this population. Enhancing the accessibility, acceptability, and effectiveness of HIV testing services for TGW in Ghanaian slums requires a multifaceted approach that integrates the principles of the gender-affirmative model. These approaches include comprehensive training for healthcare providers, the creation of inclusive and affirming healthcare environments, and the provision of culturally competent, respectful care that addresses the unique needs and challenges faced by TGW. By adopting these strategies, policymakers, healthcare providers, and community organizations can contribute to reducing health disparities and improving the overall wellbeing of TGW in Ghanaian slums.

## Supporting information

**S1 Data. Dataset of participant responses in the study on HIV testing experiences among transgender women in Ghanaian urban slums.** This dataset includes anonymized data on

positive and negative experiences, healthcare interactions, and demographic details, collected during face-to-face interviews.
(DOCX)

## Author Contributions

**Conceptualization:** Henry Delali Dakpui, Osman Wumpini Shamrock, Gloria Aidoo-Frimpong, Edem Yaw Zigah, George Rudolph Kofi Agbemedu, Abdallah Ahmed, Umar Osman, Aliyu Haruna, Gamji Rabiu Abu-Ba'are.

**Data curation:** Henry Delali Dakpui, Osman Wumpini Shamrock, George Rudolph Kofi Agbemedu, Gamji Rabiu Abu-Ba'are.

**Formal analysis:** Henry Delali Dakpui, Osman Wumpini Shamrock, Gloria Aidoo-Frimpong, Umar Osman.

**Funding acquisition:** Osman Wumpini Shamrock, Gamji Rabiu Abu-Ba'are.

**Investigation:** Henry Delali Dakpui, Osman Wumpini Shamrock, Edem Yaw Zigah, Umar Osman.

**Methodology:** Henry Delali Dakpui, Osman Wumpini Shamrock, George Rudolph Kofi Agbemedu, Abdallah Ahmed.

**Supervision:** Henry Delali Dakpui, Osman Wumpini Shamrock, Gamji Rabiu Abu-Ba'are.

**Visualization:** George Rudolph Kofi Agbemedu, Gamji Rabiu Abu-Ba'are.

**Writing – original draft:** Henry Delali Dakpui, Osman Wumpini Shamrock, Gloria Aidoo-Frimpong, Edem Yaw Zigah, George Rudolph Kofi Agbemedu, Abdallah Ahmed, Umar Osman, Aliyu Haruna, Gamji Rabiu Abu-Ba'are.

**Writing – review & editing:** Henry Delali Dakpui, Osman Wumpini Shamrock, Gloria Aidoo-Frimpong, Edem Yaw Zigah, George Rudolph Kofi Agbemedu, Abdallah Ahmed, Umar Osman, Aliyu Haruna, Gamji Rabiu Abu-Ba'are.

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
