## [Decision Letter · Decision Letter 0]

19 Aug 2024

PGPH-D-24-01260

A qualitative description of HIV testing and healthcare experiences among trans women in Ghanaian urban slums BSGH-011

Dear Dr. Shamrock,

Thank you for submitting your manuscript to PLOS Global Public Health. After careful consideration, we feel that it has merit but does not fully meet PLOS Global Public Health’s publication criteria as it currently stands. Therefore, we invite you to submit a revised version of the manuscript that addresses the points raised during the review process.

Your submission has been assessed by two reviewers, and their comments are available below. The main requests are to provide additional information about the study setting and the gender-affirmative model, as well as to enhance the Methods and Results reporting, particularly with respect to details of recruitment. Could you please also be sure to provide a copy of the interview guide?

We look forward to receiving your revised manuscript.

Kind regards,

Marianne Clemence

Staff Editor

Journal Requirements:

Additional Editor Comments (if provided):

Reviewers' comments:

Reviewer's Responses to Questions

**Comments to the Author**

1. Does this manuscript meet PLOS Global Public Health’s publication criteria? Is the manuscript technically sound, and do the data support the conclusions? The manuscript must describe methodologically and ethically rigorous research with conclusions that are appropriately drawn based on the data presented.

Reviewer #1: Yes

Reviewer #2: Yes

2. Has the statistical analysis been performed appropriately and rigorously?

Reviewer #1: N/A

Reviewer #2: Yes

3. Have the authors made all data underlying the findings in their manuscript fully available (please refer to the Data Availability Statement at the start of the manuscript PDF file)?

Reviewer #1: Yes

Reviewer #2: Yes

4. Is the manuscript presented in an intelligible fashion and written in standard English?

Reviewer #1: Yes

Reviewer #2: Yes

5. Review Comments to the Author

Reviewer #1: Review

PLOS Global Public Health - A qualitative description of HIV testing and healthcare experiences among trans women in Ghanaian urban slums BSGH-011

The study addresses a very important topic for public health, since identifying barriers to accessing HIV testing among trans women can contribute to reducing the high prevalence of HIV found in this population. However, I forward some suggestions for its improvement, with a view to future publication.

Minor issues

• The version that was submitted still had some edits to the text marked by the word change control. As this is the first submission, this should not have happened, as it could confuse the reviewer.

• To distinguish the titles of each section from the subtitles, I suggest that they be presented in capital letters in bold (INTRODUCTION, METHODS, RESULTS, DISCUSSION, CONCLUSION) and the subtitles remain in lower case letters in bold. Or, leave the subtitles in italics and without bold. I recommend checking whether the magazine's instructions have specific guidance regarding this. Otherwise, check the articles already published in this magazine to see how the titles and subtitles of each section are presented.

• I suggest including the term ‘transgender women’ in the first paragraph of the introduction, along with the acronym (TGW). This acronym only appears on page 2 (third paragraph). If you prefer not to use an acronym, I suggest standardizing the term ‘trans women’ throughout the manuscript.

• The same suggestion applies to the acronym LGBT – spell it out the first time it appears and standardize the same acronym throughout the text. It currently appears in two forms: LBGT | LGBTQ+.

• Men who have sex with men appears in full in the Discussion when an acronym for this term (MSM) has already been included in previous sections.

INTRODUCTION

It is well described and linked, however, to make the scenario related to the study population in Ghana more evident to the reader, it would be interesting to inform:

• Is there demographic data on the number of trans people living in the country?

• Are there more up-to-date epidemiological data to insert to replace reference number 2, which is from 2019? Is there a notification system to register new HIV cases, according to gender identity? Does the Ghanaian Ministry of Health prepare and make available epidemiological bulletins on HIV, which include the trans population?

• How are health services distributed in Ghana? Are they public or proven? In what type of health service is it possible to carry out an HIV test in the country? Is there a specific network of services to provide Comprehensive Health for Trans People? Are these services characterized by being trans-friendly based on any specific training of health professionals? ‘Not trans-friendly’ health services are mentioned – what criteria are used for this definition?

• As for the subtitle 'Theoretical framework', I think it could be removed, however when mentioning the gender-affirmative model, it would be interesting to provide more information about it and how its principles connect with the objective of expanding the search for health services for testing of HIV.

METHODS

• Research design – I suggest inserting the bibliographic reference on qualitative research.

• Sampling and Recruitment procedure – the use of the snowball sampling approach was mentioned. It is necessary to better explain how this technique was applied in practice. Was a map of existing slums in Accra used? Were participants from different favelas or just one? Was there already a previous connection between them and the researchers? Was a prior questionnaire administered to confirm the eligibility criteria? Inform whether the participants received financial compensation to cover transportation and food costs and what was the amount?

• Data collection procedure – insert bibliographic references when mentioning the saturation issue to define the number of participants.

• Nature of questions - Inform the content of the Guide prepared and whether there was a script to support the interviewers and which questions made up this script, in addition to the examples cited.

• Analytical strategy – explain more about the ‘meticulous approach’ and insert bibliographic references.

RESULTS

• The findings are very interesting and their description would benefit if an additional synthetic table was presented with the identified categories and subcategories.

• The first paragraph of the subitems Positive and Negative - Experiences with HIV Testing are unnecessary, as they are already included in the Description of Categories and Subcategories.

• In relation to the three people who already had HIV, it would be important to know how long ago they became aware of their status, as the quality of care in the health services used may have changed during this period. In this sense, would it be appropriate to address issues relating to access barriers and quality of care separately for these participants? I suggest saying something about it in the Discussion section.

• When presenting excerpts of the speeches, it is not necessary to identify each of them as trans women, as this does not distinguish them. Only one was identified as a ‘Muslim’, wouldn’t that make her identification easier? There are also two Christian participants aged 26, they are also indistinguishable. It would be necessary to create another form of identification.

DISCUSSION

The main findings were discussed in light of the literature and compared with the results of other research. Many points of similarity were found. There are some results that emerged in contrast to the literature. If so, it would be important to highlight it as an additional contribution of this study.

However, I missed discussing these data considering the reality of health services in terms of management. Are there government initiatives that are already underway and that involve health services? Are there government publications that explain what has been planned or is already underway to reduce the problems identified?

Every time the gender affirmation model is cited, its respective reference must be inserted. It would also be necessary to insert bibliographic references in the fourth paragraph (open communication).

Strengths and Limitations – should come before the Conclusion section

Reviewer #2: This is a well-written paper about barriers and facilitators to HIV testing among transgender women who reside in slums in the greater Accra area. The paper is interesting and informative, and I appreciate the opportunity to review it. I have only a few recommendations for improvement:

The Introduction is very comprehensive and well-written. The authors may wish to include somewhere in the introduction or in the results a Figure depicting the gender affirmative model and how it was used to guide the theoretical framework for this paper. It could be a good addition for readers to have a visual of how each facilitator and barrier identified fit into this model, if you agree this makes sense.

On page 9, it seems there is a quote missing or some editing that needs to happen as the sentence ends with “states.” For both the sections on Inclusive, Informative and Supportive Counseling and the section on Fear and Anxiety, it would be useful to break up the descriptive paragraphs and put each sentence before the corresponding quote. I think this would make the sections read more smoothly.

The main addition that could greatly improve the paper is if the authors included more information about the testing sites (not sure if this information was gathered, but if so, it would be very useful). For example, at testing sites where participants had positive experiences, was there anything they had in common contrasted with those where participants had negative experiences? One participant makes reference to the different care she received at an Outpatient unit. I am wondering if people go to a local community testing center versus a larger hospital, etc if the care they receive is different. This could also inform policies and guidelines for testing based on type of facility.

Finally, I wonder about the availability of self-testing in Ghana. Is it available at all? If so, could that be another option for those who describe that the atmosphere of testing sites and witnessing others get their results increases fear and anxiety?

6. PLOS authors have the option to publish the peer review history of their article (what does this mean?). If published, this will include your full peer review and any attached files.

**Do you want your identity to be public for this peer review?** For information about this choice, including consent withdrawal, please see our Privacy Policy.

Reviewer #1: **Yes: **Katia Cristina Bassichetto

Reviewer #2: **Yes: **Rebecca Giguere

---

## [Decision Letter · Decision Letter 1]

7 Nov 2024

A qualitative description of HIV testing and healthcare experiences among trans women in Ghanaian urban slums BSGH-012

PGPH-D-24-01260R1

Dear Dr Shamrock,

We are pleased to inform you that your manuscript 'A qualitative description of HIV testing and healthcare experiences among trans women in Ghanaian urban slums BSGH-012' has been provisionally accepted for publication in PLOS Global Public Health.

Best regards,

Julia Robinson

Executive Editor

Reviewer Comments (if any, and for reference):

Reviewer's Responses to Questions

**Comments to the Author**

1. If the authors have adequately addressed your comments raised in a previous round of review and you feel that this manuscript is now acceptable for publication, you may indicate that here to bypass the “Comments to the Author” section, enter your conflict of interest statement in the “Confidential to Editor” section, and submit your "Accept" recommendation.

Reviewer #1: All comments have been addressed

Reviewer #2: All comments have been addressed

2. Does this manuscript meet PLOS Global Public Health’s publication criteria? Is the manuscript technically sound, and do the data support the conclusions? The manuscript must describe methodologically and ethically rigorous research with conclusions that are appropriately drawn based on the data presented.

Reviewer #1: Yes

Reviewer #2: Yes

3. Has the statistical analysis been performed appropriately and rigorously?

Reviewer #1: N/A

Reviewer #2: I don't know

4. Have the authors made all data underlying the findings in their manuscript fully available (please refer to the Data Availability Statement at the start of the manuscript PDF file)?

Reviewer #1: Yes

Reviewer #2: Yes

5. Is the manuscript presented in an intelligible fashion and written in standard English?

Reviewer #1: Yes

Reviewer #2: Yes

6. Review Comments to the Author

Reviewer #1: I reviewed the submitted version and consider that all suggestions were accepted in the current version. As my only observation, I missed the mention of table 1 in the text. Correct the term, it is not table but chart.

Reviewer #2: Thank you for your work on this revision. The paper is much improved.

7. PLOS authors have the option to publish the peer review history of their article (what does this mean?). If published, this will include your full peer review and any attached files.

**Do you want your identity to be public for this peer review?** For information about this choice, including consent withdrawal, please see our Privacy Policy.

Reviewer #1: No

Reviewer #2: **Yes: **Rebecca Giguere
